# Enhancing Conversational Recommender Systems with Tree-Structured Knowledge and Pretrained Language Models

## Abstract

Conversational recommender systems (CRS) have emerged as a key enhancement to traditional recommendation systems, offering interactive and explainable recommendations through natural dialogue. Recent advancements in pretrained language models (PLMs) have significantly improved the conversational capabilities of CRS, enabling more fluent and context-aware interactions. However, PLMs still face challenges, including hallucinations—where the generated content can be factually inaccurate—and difficulties in providing precise, entity-specific recommendations. To address these challenges, we propose the PCRS-TKA framework, which integrates PLMs with knowledge graphs (KGs) through prompt-based learning. By incorporating tree-structured knowledge from KGs, our framework grounds the PLM in factual information, thereby enhancing the accuracy and reliability of the recommendations. Additionally, we design a user preference extraction module to improve the personalization of recommendations and introduce an alignment module to ensure semantic consistency between dialogue text and KG data. Extensive experiments demonstrate that PCRS-TKA outperforms existing methods in both recommendation accuracy and conversational fluency. The code is anonymously open-sourced at https://anonymous.4open.science/r/PCRS-TKA-9496.

## 1 Introduction

Recommendation systems are pivotal in intelligent assistants, helping users discover relevant items more efficiently. However, traditional recommendation systems typically focus on item suggestions without interactive dialogues with users (Chen et al., 2017; He et al., 2020). To address this limitation, conversational recommender systems (CRS) have gained significant attention in recent years, which enhance not only the flexibility of the recommendations but also their explainability by allowing more natural and intuitive user interactions (Christakopoulou et al., 2016; Tran et al., 2020). The recent advancements in pretrained language models (PLMs) have significantly expanded the capabilities of CRS with their powerful language understanding and generation abilities (Zhang & Wang, 2023; Wu et al., 2021). However, PLMs face their own set of challenges in recommendation tasks, particularly hallucination, where the models may generate factually incorrect information (Hal; Zhang et al., 2023). To counter this, knowledge graphs (KGs) can be used as auxiliary tools, grounding PLMs in factual external knowledge (Wang et al., 2022b; Tong et al., 2024), thus enhancing both the accuracy and reliability of CRS.

In the literature, conventional CRS primarily relied on structured conversations centered around item attributes such as genre or price (Gao et al., 2021a). Later, KG-based CRS (Wang et al., 2019; Petroni et al., 2019; Bouraoui et al., 2020) incorporated external knowledge resources and developed specialized alignment strategies to ensure semantic consistency. For instance, KBRD (Chen et al., 2019) combined KGs with relational graph convolutional networks (RGCNs) (Schlichtkrull et al., 2017) to enhance interaction between recommendation and dialogue modules. KGSF (Zhou et al., 2020a) extended this by incorporating word-level KGs and using mutual information maximization (MIM) to align word and entity representations. RevCore (Lu et al., 2021) enriched dialogues by incorporating unstructured review data for more diverse responses, while C2CRS (Zhou et al., 2023) employed multi-granularity contrastive learning to align multimodal data and improve semantic consistency. Despite these advancements, natural language capabilities remained limited. More recent

works, such as BARCOR (Wang et al., 2022a) and UniCRS (Wang et al., 2022b), have adopted pre-trained language models (PLMs) and prompt learning (Chen et al., 2022) to generate higher-quality conversational responses. These PLM-based models also integrate KGs to address hallucination issues and improve domain-specific knowledge.

However, several key challenges remain in integrating PLMs and KGs for conversational recommendation. First, existing methods often rely on graph convolutional networks (GCNs), such as RGCN, to extract relational information from KGs. While effective, these approaches do not fully exploit the reasoning capabilities of PLMs over graph relationships, thereby limiting the potential of CRS for more sophisticated knowledge integration. Second, dialogue text in current methods is often treated merely as textual input data, neglecting valuable latent user collaborative preference information embedded in conversations. This oversight can result in suboptimal personalized recommendations that fail to align with users' true preferences. Third, CRS typically involve multiple data types, such as textual dialogue data and structured knowledge from graphs, which reside in distinct semantic spaces (Li et al., 2021). Without proper semantic alignment, the integration of these heterogeneous data sources may introduce noise, ultimately degrading recommendation quality.

In response to these challenges, we propose a novel framework, PCRS-TKA (Prompt-based Conversational Recommender System with Tree-structured Knowledge Augmentation), that effectively integrates KG and dialogue data through prompt-based learning using a PLM. Specifically, our approach first extracts a knowledge tree from the KG based on historically interacted items, leveraging the PLMs's logical reasoning abilities to capture entity relationships. Additionally, we design a user preference extraction module that captures user collaborative preferences through dialogue interaction, guiding more personalized recommendations. Then, our framework introduces an alignment module that harmonizes data from different semantic spaces, minimizing noise and improving recommendation accuracy. Finally, through extensive experiments on public datasets, we demonstrate that PCRS-TKA outperforms existing methods on both the recommendation and conversation tasks.

## 2 RELATED WORK

The related work can be categorized into two main areas: **Conversational Recommendation Systems** and **Combining PLMs and KGs for Recommendation**.

### 2.1 CONVERSATIONAL RECOMMENDATION SYSTEMS

Early CRS methods (Chen et al., 2019; 2017) focused on structured conversations that gathered user preferences through item attributes like genre or price, relying on predefined templates and algorithms such as multi-armed bandits or reinforcement learning. However, these approaches lacked flexibility and natural language generation capabilities. To improve this, KG-based CRS methods were developed. KBRD (Chen et al., 2019) introduced KGs and RGCNs to better connect recommendation and dialogue tasks by modeling complex relations between items and users. KGSF (Zhou et al., 2020a) extended this by incorporating word-level KGs and using MIM to align word and entity representations, resulting in more coherent responses. RevCore (Lu et al., 2021) enriched dialogue generation by leveraging unstructured review data, adding diversity to system responses, while C2CRS (Zhou et al., 2023) used multi-granularity contrastive learning to align multimodal data (e.g., text, images, KG). Despite these advancements, KG-based methods often treated recommendation and dialogue modules separately, limiting their ability to fully utilize dialogue content. With the rise of PLMs, prompt learning has been introduced to improve conversational capabilities. For example, BARCOR (Wang et al., 2022a) employs BART (Lewis et al., 2020) to generate higher-quality responses, while UniCRS (Wang et al., 2022b) uses prompt learning to integrate recommendation and dialogue generation. These PLM-based systems also incorporate KGs to mitigate hallucination issues and improve domain-specific knowledge, providing better alignment between dialogue and recommendations.

### 2.2 COMBINING PLMS AND KGS FOR RECOMMENDATION

Recommendation tasks often require systems to have prior knowledge of the domain of the recommended entities, a capability that PLMs typically lack. KGs can compensate for this deficiency by providing structured, domain-specific knowledge. To integrate KGs, many approaches (Li et al., 2021; Gao et al., 2021a) align the semantic space of PLMs with the KG to obtain more accurate user feature representations. This is commonly achieved by modifying the Transformer architecture with cross-attention mechanisms, enabling the model to process both dialogue text and KG information simultaneously. For instance, works like KGSF (Zhou et al., 2020a) use mutual information maxi-

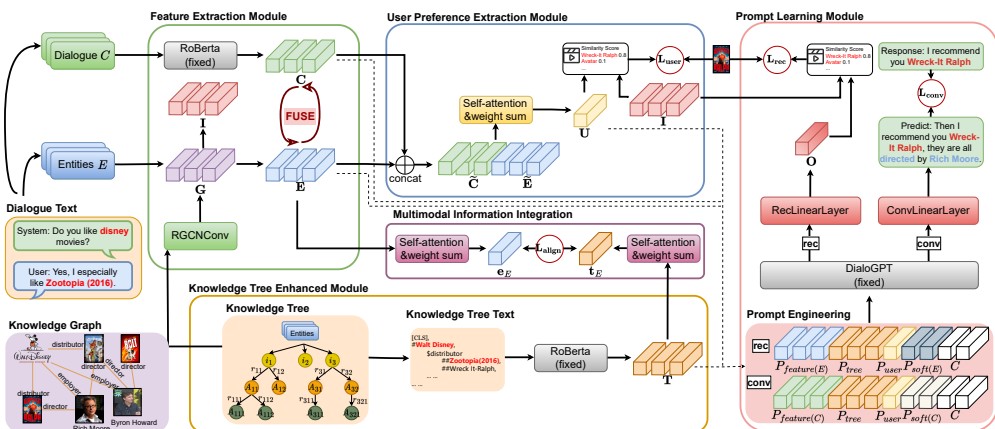

Figure 1: The network architecture of the PCRS-TKA framework.

mization to align the representations of PLMs and KGs, while C2CRS (Zhou et al., 2023) employs contrastive learning to align these spaces at both the sentence and word levels. These methods have shown promising results. However, in prompt learning methods, the structure of pre-trained LLMs remains fixed, as their parameters are optimized on large text corpora. Therefore, methods like UniCRS(Wang et al., 2022b) incorporate KG information by concatenating implicit vectors from graph neural networks (GNNs) into the input prompts. While this approach helps integrate KG data, it does not fully leverage the reasoning capabilities of the PLM in understanding the relationships within the KG.

Different from the above works, we aim to fully leverage both the reasoning capabilities of PLMs and the structured knowledge within KGs. We integrate KG information into the prompt-based learning process with a novel knowledge tree-enhanced module combined with a user preference extraction module, which helps the system provide more accurate and context-aware recommendations.

## 3 METHODOLOGY

In this section, we will first introduce our task formulation and the overview of the proposed PCRS-TKA framework. Then we will provide the technical details of our proposed framework.

### 3.1 TASK FORMULATION

The goal of CRS is to recommend relevant items while engaging in a natural, ongoing conversation with the user. During each turn of the conversation, the system analyzes the dialogue history, infers the user's preferences, and generates a response that includes recommended items within the natural language utterance. If the recommended items do not meet the user's needs, the system continues the conversation in subsequent turns, refining the recommendations based on the user's feedback and evolving preferences.

Formally, let $u$ denotes a user from user set $\mathcal{U}$, $i$ denotes an item from item set $\mathcal{I}$, and $w$ denotes a word from vocabulary $\mathcal{V}$. A multi-turn conversation $C$ consists of a set of utterances, denoted by $C = \{s_t\}_{t=1}^n$, in which $s_t$ denotes the sentence in the $t$-th turn and composed of words from the vocabulary $\mathcal{V}$, denoted as $s_t = \{w_j\}_{j=1}^m$. For a given $n$-turn conversation $C$, the task of CRS is to generate a response utterance $s_{n+1}$ and select a set of recommended items $I_{n+1}$ from the item set $\mathcal{I}$, where $I_{n+1}$ can be empty. We denote the exterenal KG as $G$. It stores a semantic fact with a triple $< e_1, r, e_2 >$, where $e_1, e_2$ from the entity set $\mathcal{E}$ and $r$ from the relation set $\mathcal{R}$, here we assume all the candidate items can be find in $\mathcal{E}$, i.e.$\mathcal{I} \subset \mathcal{E}$.

### 3.2 OVERVIEW OF THE APPROACH

As shown in Figure 1, we propose the **PCRS-TKA** framework for CRS, which integrates PLMs and KGs through knowledge-enhanced prompt learning. For a given dialogue text $C$, we previously extract the KG entities that appear in $C$, and denote as $E$. Then the framework consists of four main components: **(1) Feature Encoder Module:** We employ RoBERTa (Liu et al., 2019) and RGCN (Schlichtkrull et al., 2017) to encode dialogue text $C$ and the KG $G$. The dialogue embeddings $\mathbf{C}$ and entity embeddings $\mathbf{E}$ (extracted from $G$) are fused to form the first part of the prompt, denoted

as $P_{\text{feature}}$. $\mathbf{E}$ is used for recommendation, while $\mathbf{C}$ is used for dialogue generation. **(2) Knowledge Tree Enhanced Module:** We transform the KG triples related to entities $E$ into a knowledge tree, which is then converted into a long text $T$. RoBERTa encodes this text, and the resulting embeddings $\mathbf{T}$ are aligned with $\mathbf{E}$ to form the second part of the prompt, $P_{\text{tree}}$. **(3) User Preference Extraction Module:** This module infers user preferences through an auxiliary recommendation task, generating a preference matrix $\mathbf{U}$. Fused with $\mathbf{E}$, this forms the third part of the prompt, $P_{\text{user}}$. **(4) Soft-Prompt Module:** A learnable soft prompt $P_{\text{soft}}$ is appended to the end of the prompt for additional task-specific guidance. Since the PLM remains fixed during prompt learning, our focus is on constructing a knowledge-enhanced prompt template to integrate different information. Finally, all components of the prompt ($P_{\text{feature}}, P_{\text{tree}}, P_{\text{user}}, P_{\text{soft}}$) are concatenated with the dialogue text $C$ and fed into DialoGPT for conversation and recommendation tasks.

### 3.3 FEATURE ENCODER MODULE

For natural language understanding, we adopt RoBERTa (Liu et al., 2019), a bidirectional pre-trained language model, to encode the dialogue text $C$ and generate word embedding matrix $\mathbf{C}$, where $\mathbf{C} \in \mathbb{R}^{n_W \times d_W}$, $n_W$ is the number of words in the dialogue text and $d_W$ the hidden size of RoBERTa. RoBERTa excels at capturing the contextual meaning of words, making it suitable for understanding user intents from conversation.

Furthermore, since pre-trained language models lack domain-specific knowledge, we incorporate a KG $G$ to provide external knowledge about entities mentioned in the conversation. KGs contain rich relational information. To encode this information, we use RGCN, which learns the representations of entities in KGs. By aggregating information from neighboring nodes, we obtain the embedding matrix of all entities in $G$, and denote it as $\mathbf{G}$, where $\mathbf{G} \in \mathbb{R}^{n_G \times d_E}$, $n_G$ is the number of entities in $G$, i.e. the size of the entity set $\mathcal{E}$, $n_E$ is the hidden size of RGCN. The entity embeddings corresponding to $E$ (i.e., entities mentioned in $C$) are retrieved from matrix $\mathbf{G}$ and denoted as $\mathbf{E}$, , where $\mathbf{E} \in \mathbb{R}^{n_E \times d_E}$, $n_E$ is the number of entities in $E$.

Dialogue text typically captures user preferences and intents, while entity embeddings from the R-GCN provide external knowledge about entities. However, there is a significant semantic gap between the two modalities. To bridge this gap, we apply a cross-interaction mechanism (Wang et al., 2022b), which aligns the semantic spaces of the word and entity embeddings:

$$\mathbf{C} = \mathbf{C}W_C, \qquad \mathbf{E} = \mathbf{E}W_E, \qquad \mathbf{A} = \mathbf{C}\mathbf{W}\mathbf{E}^\top, \qquad \widetilde{\mathbf{C}} = \mathbf{C} + \mathbf{E}\mathbf{A}, \qquad \widetilde{\mathbf{E}} = \mathbf{E} + \mathbf{C}\mathbf{A}^\top, \qquad (1)$$

where $W_C \in \mathbb{R}^{d_C \times d}, W_E \in \mathbb{R}^{d_E \times d}, W \in \mathbb{R}^{d \times d}$ are learnable weight matrix. Through this bilinear transformation, the model captures the correlations between dialogue text and entities, producing refined representations $\widetilde{\mathbf{C}} \in \mathbb{R}^{n_C \times d}$ and $\widetilde{\mathbf{E}} \in \mathbb{R}^{n_E \times d}$ that share information from both modalities.

### 3.4 KNOWLEDGE TREE ENHANCED MODULE

To fully leverage the information from the KG $G$, we propose to integrate KG entities into the original dialogue context by constructing a tree-structured knowledge tree. This structure, as illustrated in Figure 2, enhances the dialogue by embedding relevant background knowledge, taking advantage of PLM's ability to process and analyze complex semantic structures.

Specifically, given the dialogue $C$ and the mentioned entities $E$, we begin by selecting these entities as seed nodes and extract their n-hop triples from the KG $G$. These triples are then reorganized into $t$ n-layer trees. To unify these $t$ trees, we introduce a common parent node, constructing a comprehensive knowledge tree $G_{\text{tree}}$ corresponding to the entities in $E$. To preserve the relational structure of $G_{\text{tree}}$, we represent the edges between consecutive layers of nodes as virtual nodes. By performing a depth-first traversal of the entire tree, we generate an ordered list of nodes in the form of "#Moana (2016)", where "Moana (2016)" is the text of the node, and "#" represents the node's depth in the tree (e.g., depth 1). This depth marker ensures that the tree structure can be fully preserved and easily reconstructed from the text list. Finally, we concat the node text in the pairs in order to get the equivalent text of the knowledge tree, and input them to the RoBerta to get the embedding matrix $\mathbf{T}$ for the knowledge tree, where $\mathbf{T} \in \mathbb{R}^{n_T \times d_W}$, $n_T$ is the length of the knowledge tree text.

### 3.5 USER PREFERENCE EXTRACTION MODULE

In multi-turn dialogues, the user's preferences for candidate items in the last turn significantly influence the recommendations in the current turn. Hence, we design a novel module to capture user

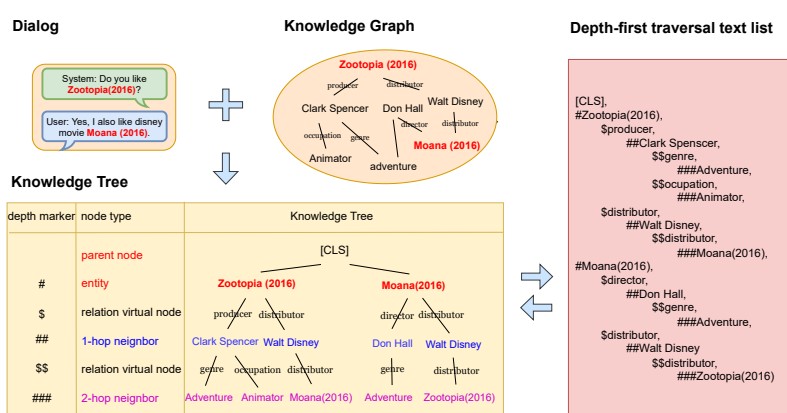

Figure 2: The example of constructing a knowledge tree.

preferences from the multi-turn dialogue $C$, and use this information to guide the training of the PLM with collaborative signals.

Given the fused entity embeddings $\widetilde{\mathbf{E}}$ from RGCN and the fused word embeddings $\widetilde{\mathbf{C}}$ from RoBERTa, we concatenate them and apply a self-attention mechanism to capture relationships between all elements in the sequence. The self-attention operation helps compute a weighted sum of the embeddings, resulting in the user's preference embedding $\mathbf{U}$, where $\mathbf{U} \in \mathbb{R}^d$:

$$\mathbf{X} = \text{concat}(\widetilde{\mathbf{C}}, \widetilde{\mathbf{E}}), \tag{2}$$

$$\mathbf{U} = \sum_{i=1}^{n} \sum_{j=1}^{n} \text{softmax} \left( \frac{\mathbf{X}_i W_Q \cdot (\mathbf{X}_j W_K)^\top}{\sqrt{d}} \right) \mathbf{X}_j W_V, \tag{3}$$

where $W_Q \in \mathbb{R}^{d \times d}, W_K \in \mathbb{R}^{d \times d}, W_V \in \mathbb{R}^{d \times d}$ are learnable weight matrices.

Next, we introduce an auxiliary task to enhance the module's ability to capture collaborative user preference information. Given the learned user preference embedding $\mathbf{U}$ and item embeddings $\mathbf{I} \in \mathbb{R}^{n_I \times d_E}$ ($\mathbf{I}$ consists of the embeddings of the candidate items in $\mathcal{I}$ within the graph entity embeddings $\mathbf{G}$), we compute the rating score for each item $i$ in the candidate item set $\mathcal{I}$ as follows:

$$\mathbf{I} = \mathbf{I} W_E, \quad R = \text{softmax}(\mathbf{U}^\top \mathbf{I}), \tag{4}$$

where $W_E$ is a learnalble weight matrix to transform the dommension of $\mathbf{I}$ from $d\_E$ to $d$ to consistent with $\mathbf{U}$, $R \in [0,1]^{1 \times n_I}$ is the predicted rating score of user over candidate items. Similarly, given $N$ conversations and $\mathbf{Y} \in \{0,1\}^{N \times M}$ represents the ground truth rating score, we can compute the predicted rating score $R \in \{0,1\}^{N \times n_I}$ in all conversations and give the following cross-entropy loss(Mao et al., 2023):

$$L_{user} = -\sum_{j=1}^{N} \sum_{i=1}^{n_I} \left[ Y_j^i \cdot \log R_j^i + (1 - Y_j^i) \cdot \log(1 - R_j^i) \right]. \tag{5}$$

## 3.6 MULTIMODAL INFORMATION INTEGRATION

Our framework utilizes multimodal information from both conversations and KG. Effectively processing and integrating these modalities is essential to fully exploit the available information.

As introduced in Section 3.3, when aligning entity embeddings $\mathbf{E}$ and conversation embeddings $\mathbf{C}$, some degree of information loss is inevitable. For instance, in dialogue tasks, conversational information is more critical, while in recommendation tasks, KG information plays a larger role. Therefore, we previously introduced a cross-interaction module to obtain two embeddings $\widetilde{\mathbf{E}}$ and $\widetilde{\mathbf{C}}$, which are used for subsequent tasks.

In contrast, both entity embeddings $\mathbf{E}$ and knowledge tree embeddings $\mathbf{T}$ are derived from the KG and represent the same underlying information. Aligning these embeddings helps bring closer the

representations of the same entity in the semantic space. Specifically, for the entities $E$ mentioned in the conversation, we perform a self-attention (Vaswani et al., 2023) and weighted summation operation separately on $\mathbf{E} = [e_1; e_2; ...e_{n_E}]$ and $\mathbf{T} = [t_1; t_2; ...t_{n_T}]$, obtaining two views of $E$:

$$\mathbf{e}_E = \sum_{i=1}^{n_E} \sum_{j=1}^{n_E} \text{softmax}\left( \frac{e_i W_Q' \cdot (e_j W_K')^\top}{\sqrt{d}} \right) e_j W_V', \tag{6}$$

$$\mathbf{t}_E = \sum_{i=1}^{n_T} \sum_{j=1}^{n_T} \text{softmax}\left( \frac{t_i W_Q'' \cdot (t_j W_K'')^\top}{\sqrt{d}} \right) t_j W_V'', \tag{7}$$

The resulting representations $\mathbf{e}_E \in \mathbb{R}^{1 \times d}$ and $\mathbf{t}_E \in \mathbb{R}^{1 \times d}$ should be aligned in the semantic space. To achieve this, we apply contrastive learning(Ma & Collins, 2018) to minimize the semantic distance between positive pairs while maximizing the distance between negative pairs. Specifically, for $\mathbf{e}_{E_i}, \mathbf{t}_{E_i}, \mathbf{e}_{E_j}, \mathbf{t}_{E_j}$ extracted from two entitiy sequences $E_i$ and $E_j$, two pairs $(\mathbf{e}_{E_i}, \mathbf{t}_{E_j})$ and $(\mathbf{t}_{E_i}, \mathbf{e}_{E_j})$ are treated as positive pairs $E_i$ equals to $E_j$ and negative pairs otherwise. Let $M_{i,j} = 1, \text{if } E_i == E_j, \text{otherwise} M_{i,j} = 0$. For a mini-batch with $b$ conversations, which means there are $b$ entity sequences $[E_1, E_2...E_b]$, the contrastive learning loss is formulated as:

$$L_{\text{align}} = \sum_{i=1}^{b} \sum_{j=1}^{b} -\log \frac{\exp(\mathbf{e}_{E_i} \cdot \mathbf{t}_{E_j}/\tau)}{\sum_{k=1}^{b}(\exp(\mathbf{t}_{E_i} \cdot \mathbf{e}_{E_k}/\tau) + \exp(\mathbf{e}_{E_i} \cdot \mathbf{t}_{E_k}/\tau)) \cdot M_{i,k}} \cdot M_{i,j}, \tag{8}$$

where $\tau$ is a temperature hyperparameter.

### 3.7 PROMPT LEARNING MODULE

The model parameters are divided into four groups: the base PLM, feature encoder module, user preference extraction module, and task-specific soft tokens, denoted as $\Theta_{plm}$, $\Theta_{feature}$, $\Theta_{user}$, and $\Theta_{soft}$, respectively. We adopt DialoGPT (Zhang et al., 2019), a Transformer-based autoregressive model pre-trained on large-scale dialogue data from Reddit, as the base PLM. DialoGPT generates contextual representations from input tokens and is well-suited for recommendation systems (Wang et al., 2022b). Importantly, the parameters of $\Theta_{plm}$ are kept fixed during training, while we optimize the other parameters.

For the conversation task, we use $\widetilde{\mathbf{C}}$ as the first part of the prompt, denoted as $P_{\text{feature(C)}}$, and for the recommendation task, we use $\widetilde{\mathbf{E}}$, denoted as $P_{\text{feature(E)}}$. Next, knowledge tree embeddings $\mathbf{T}$ are generated via the knowledge tree module, forming the second part of the prompt, $P_{\text{tree}}$. The user's preference embedding $\widetilde{\mathbf{U}}$ is extracted and used as the third part of the prompt, $P_{\text{user}}$. Lastly, we initialize soft-prompt tokens for conversation and recommendation tasks, denoted as $P_{soft(C)}$ and $P_{soft(E)}$, respectively. The complete input to the PLM for both tasks is constructed by concatenating the prompt components with the dialogue text $C$:

$$\widetilde{C}_{gen} = \text{concat}(P_{feature(C)}, P_{tree}, P_{user}, P_{soft(C)}, C), \tag{9}$$

$$\widetilde{C}_{rec} = \text{concat}(P_{feature(E)}, P_{tree}, P_{user}, P_{soft(E)}, C). \tag{10}$$

The training for both conversation and recommendation tasks consists of two stages. For the recommendation task, in the first stage, we pre-train the parameters of $\Theta_{feature}$ and $\Theta_{user}$ based on the self-supervised response generation task. In the second stage, we randomly initialize the parameters of soft tokens $\Theta_{soft(E)}$, and learn the $\Theta_{feature}$, $\Theta_{user}$, and $\Theta_{soft(E)}$ simultaneously.

For the recommendation task, we use the following loss functions:

$$\mathbf{O} = \text{Pooling}(f(\widetilde{C}_{rec}|\Theta_{plm}; \Theta_{feature}; \Theta_{user}; \Theta_{soft(E)};)), \tag{11}$$

$$\hat{R} = \text{Softmax}(\mathbf{O} \cdot \mathbf{I}), \tag{12}$$

$$L_{rec} = -\sum_{j=1}^{N} \sum_{i=1}^{M} [Y_j^i \log \hat{R}_j^i + (1 - Y_j^i) \log(1 - \hat{R}_j^i)], \tag{13}$$

$$L_{all} = L_{rec} + \alpha L_{user} + \beta L_{align}, \tag{14}$$

where $f(\mathbf{X}|\Theta)$ denote the output of DialoGPT parameterized by $\theta_{plm}$ taking a token sequence $\mathbf{X}$ as input, the pooling operation can be chosen from averaging, max pooling, or selecting the embedding of the first token, $\alpha$ and $\beta$ are hyperparameters.

For the conversation task, we follow a similar two-stage process while using $P_{\text{feature(C)}}$ and $\Theta_{soft(C)}$ to replace $P_{\text{feature(E)}}$ and $\Theta_{soft(E)}$. Also we use the text generation loss of DialoGPT to replace $L_{rec}$ in both of the two-stage training processes.

## 4 EXPERIMENTS

In this section, we first introduce the datasets and experimental settings. Next, we present the overall evaluation of our framework, comparing it against several state-of-the-art baselines on both the recommendation and conversation tasks. Finally, we provide detailed analyses through ablation studies and sensitivity experiments for key hyperparameters. The source code and data have been uploaded to an anonymous repository: `https://anonymous.4open.science/r/PCRS-TKA-9496`.

### 4.1 EXPERIMENTAL SETUP

**Dataset.** We conducted multiple experiments on the REDIAL (Li et al., 2019) and INSPIRED (Hayati et al., 2020) datasets. The REDIAL dataset is a conversational dataset for movie recommendations, containing 10,006 dialogues with 182,150 utterances about recommendations for 6,281 movies. INSPIRED is a dataset containing 1001 dialogues with 35,811 utterances about recommendations for 1472 movies. Both datasets are constructed by Amazon Mechanical Turk (AMT). In the experiment, we split the datasets into training, validation, and test sets in an 8:1:1 ratio. For each dialogue, we incrementally added one round of utterances starting from the first round to create new data, thereby expanding the dataset.

**Knowledge Graph.** DBpedia (Auer et al., 2007) is a large-scale, multilingual KG extracted from the structured content of Wikipedia. It represents real-world entities and their relationships, such as people, places, and organizations, along with their attributes, containing 5,040,986 high-frequency entities with their corresponding 927 relations and 24,267,796 triplets. For the entire DBpedia graph is too huge, we collected all the entities appearing in the dataset corpus via the Tagme tool. Starting from these entities as seeds, we extracted their one-hop triples on the DBpedia graph, and the subgraph obtained is used as the external KG in our experiment.

**Evaluation Metrics.** We conducted two types of evaluations: recommendation evaluation and dialogue evaluation. For the recommendation task, we used recall@k (k=10, 50), NDCG@k (k=10, 50), and MRR@k (k=10, 50) as metrics. For the conversation task, we employed both automatic and manual evaluations. For automatic evaluation, we used word-level distinct-n (n=1, 2, 3, 4) to measure response diversity. Additionally, we randomly selected 100 conversations and their corresponding model-generated responses, and invited ten annotators to manually score the responses. The manual evaluation assessed three aspects: *Fluency*, *Informativeness*, and *Question-Answer Consistency*, with scores ranging from 0 to 5. The details of the manual evaluation, along with the original survey data, have been uploaded to an anonymous repository: `https://anonymous.4open.science/r/PCRS-TKA-9496`.

**Benchmark Models.** We selected several state-of-the-art recommendation models as baselines, including: **ReDial** (Li et al., 2019): This model was introduced with the Redial dataset. It integrates an autoencoder-based recommendation module and an HRED-based conversation module. **KBRD** (Chen et al., 2019): This model utilizes DBpedia to enhance the semantic representation of entities, combining a self-attention-based recommendation module with a Transformer-based conversation module. **KGSF** (Zhou et al., 2020a): It integrates ConceptNet (Speer et al., 2018) and DBpedia to enhance word and entity representations, and uses mutual information maximization to align the semantic spaces of both KGs. **TG-ReDial** (Zhou et al., 2020b): This model introduces a topic prediction task, employing SASRec (Kang & McAuley, 2018) for recommendation, a BERT (Devlin et al., 2019) encoder for topic prediction, and GPT-2 (Gao et al., 2021b) for response generation. **UniCRS** (Wang et al., 2022b): This model uses prompt learning to guide a pretrained large language model for both recommendation and conversation tasks, enhancing the semantic representation of entities with DBpedia.

**Implementation Details.** We chose the DialoGPT-small model as the base PLM for prompt learning and used the RoBERTa-base model as the encoder for text, freezing all parameters of these two modules during the entire training process. We have used grid search to choose the hyperparameters. After searching, we used AdamW with epsilon set to 0.01, learning rate set to 5e-4 for first-stage pre-training, and 1e-4 for second-stage training for both recommendation and conversation tasks. The batch size was set to 64 for the recommendation task and 8 for the conversation task. The soft

Table 1: Results on recommendation task. Numbers marked with * indicate that the improvement is statistically significant compared with the best baseline.

| Model | recall@10 | recall@50 | ndcg@10 | ndcg@50 | mrr@10 | mrr@50 |
|---|---|---|---|---|---|---|
| **Datasets** | | | **INSPIRED** | | | |
| REDIAL | 0.106 | 0.223 | 0.049 | 0.075 | 0.031 | 0.037 |
| KBRD | 0.151 | 0.278 | 0.102 | 0.128 | 0.086 | 0.091 |
| KGSF | 0.178 | 0.294 | 0.109 | 0.133 | 0.088 | 0.093 |
| TG-ReDial | 0.173 | 0.331 | 0.110 | 0.144 | 0.091 | 0.098 |
| UNICRS | 0.262 | 0.406 | 0.159 | 0.193 | 0.131 | 0.138 |
| **PCRS-TKA** | **0.273**\* | **0.445**\* | **0.184**\* | **0.220**\* | **0.156**\* | **0.162**\* |
| *Improvement (%)* | *4.20%* | *9.61%* | *15.72%* | *13.99%* | *19.08%* | *17.39%* |
| **Datasets** | | | **Redial** | | | |
| REDIAL | 0.050 | 0.186 | 0.024 | 0.053 | 0.015 | 0.021 |
| KBRD | 0.189 | 0.372 | 0.101 | 0.141 | 0.074 | 0.082 |
| KGSF | 0.177 | 0.369 | 0.094 | 0.137 | 0.069 | 0.078 |
| TG-ReDial | 0.179 | 0.353 | 0.101 | 0.140 | 0.078 | 0.086 |
| UNICRS | 0.213 | 0.414 | 0.119 | 0.163 | 0.090 | 0.100 |
| **PCRS-TKA** | **0.220**\* | **0.432**\* | **0.128**\* | **0.169**\* | **0.093**\* | **0.103**\* |
| *Improvement (%)* | *3.29%* | *4.35%* | *7.56%* | *3.68%* | *3.33%* | *3.00%* |

Table 2: Evaluation results on the conversation task. Numbers marked with * indicate that the improvement is statistically significant compared with the best baseline.

| Models | INSPIRED | | | | ReDial | | | |
|---|---|---|---|---|---|---|---|---|
| | distinct-1 | distinct-2 | distinct-3 | distinct-4 | distinct-1 | distinct-2 | distinct-3 | distinct-4 |
| ReDial | 0.036 | 0.313 | 1.237 | 2.562 | 0.010 | 0.070 | 0.279 | 0.643 |
| KBRD | 0.067 | 0.567 | 2.017 | 3.621 | 0.011 | 0.094 | 0.488 | 1.004 |
| KGSF | 0.077 | 0.657 | 2.822 | 5.992 | 0.011 | 0.110 | 0.656 | 1.729 |
| TG-ReDial | 0.087 | 0.778 | 2.825 | 5.511 | 0.232 | 1.016 | 1.487 | 1.642 |
| UniCRS | 1.404 | 3.949 | 6.004 | 7.082 | 0.307 | 0.899 | 1.267 | 1.390 |
| **PCRS-TKA** | **2.209**\* | **6.851**\* | **9.676**\* | **10.465**\* | **0.383**\* | **1.144**\* | **1.646**\* | **1.825**\* |
| *Improvement (%)* | *57.32%* | *73.53%* | *61.19%* | *47.72%* | *24.76%* | *27.25%* | *29.96%* | *31.29%* |

prompt token length was set to 10 for the recommendation task and 20 for the conversation task according to parameter tuning results. For all baseline methods, we also use the grid search for tuning hyperparameters. More details can be found in the Appendix.

## 4.2 EVALUATION ON RECOMMENDATION TASK

As shown in Table 1, our model consistently outperforms all baseline models across both the IN-SPIRED and ReDial datasets in the recommendation task. Specifically, our model achieves significant improvements over the best-performing baseline, UniCRS, with improvements of up to 19.08% on INSPIRED and 7.56% on ReDial in terms of key metrics like recall@10 and ndcg@10. It is noticed that on the INSPIRED dataset, our model achieves higher gains compared to the ReDial dataset. This may be because INSPIRED contains more dialogue than ReDial. Thus PCRS-TKA can learn complex user preferences aligned with textual information, resulting in larger performance gains. When comparing baseline models, we observe that methods utilizing external KGs (KBRD, KGSF) generally perform better than the basic ReDial model. KGSF, which integrates two external KGs (DBpedia and ConceptNet), shows better performance than KBRD, which uses only DBpedia. However, their improvement is limited compared to UniCRS, which leverages pretrained language models and prompt learning techniques. Our model further uses a tree-structured KG to enhance semantic alignment and incorporate user preference extraction mechanisms from multi-turn dialogues, so that we can better handle the complexities of the conversation context, leading to its superior performance over all baselines.

## 4.3 EVALUATION ON CONVERSATION TASK

Table 2 presents the evaluation results on the conversation task, where our model, PCRS-TKA, outperforms all baselines across both datasets on the distinct-n metrics. Specifically, our model achieves improvements of up to 73.53% in distinct-2 on the INSPIRED dataset and 31.29% in distinct-4 on the ReDial dataset compared to the best baseline, UniCRS. These results indicate that PCRS-TKA generates more diverse responses, which is crucial for maintaining engaging and natural conversations. However, it is important to note that distinct-n metrics, while useful for measuring lexical diversity, do not fully capture the complexities of dialogue quality in conversational recommender

Table 3: Human evaluation results on the conversation task. Numbers marked with * indicate that the improvement is statistically significant compared with the best baseline.

| Models | Fluency | Informativeness | Question-Answer Consistency |
|---|---|---|---|
| ReDial | 3.17 | 2.11 | 2.04 |
| KGSF | 3.08 | 1.98 | 1.96 |
| UniCRS | 3.92 | 3.36 | 3.28 |
| **PCRS-TKA** | **4.38***  | **3.94***  | **3.77***  |
| *Improvement (%)* | *11.73%* | *17.26%* | *14.94%* |

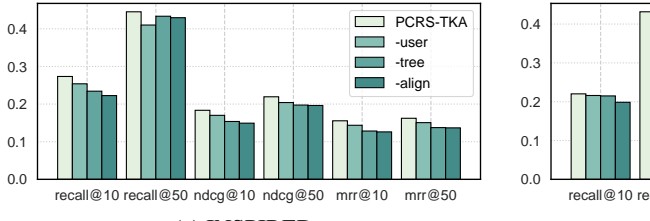
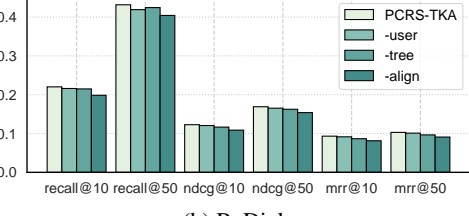

(a) INSPIRED                                       (b) ReDial

Figure 3: Ablation study on both two datasets about the recommendation task. User and tree refer to two kinds of prompts. Align refers to the information alignment module in contrastive loss.

systems. Table 3 shows the results of the human evaluation, where PCRS-TKA significantly outperforms all baselines in terms of fluency, informativeness, and question-answer consistency. Our model achieves 11.73% improvement in fluency, 17.26% in informativeness, and 14.94% in Q&A consistency over UniCRS, the strongest baseline. The superior performance of PCRS-TKA is driven by two main factors. First, our model integrates structured knowledge from the KG into the prompt, enriching the dialogue with contextually relevant information, thereby improving fluency and informativeness. Second, our model extracts and leverages user preferences across multi-turn conversations, ensuring responses align with the user's evolving needs, leading to better Q&A consistency. In comparison, UniCRS outperforms KGSF and ReDial due to its use of PLMs, which excel in text generation and comprehension. KGSF benefits from external KGs but cannot fully capture conversational dynamics and user preferences as PCRS-TKA does.

### 4.4 ABLATION STUDY

**Recommendation Task.** We performed ablation experiments to evaluate the contribution of each component in our approach across both datasets. Specifically, we excluded the knowledge tree-enhanced prompts, user preference prompts, and the information alignment module, separately during both the pre-training and training phases. As shown in Figure 3, removing any component leads to performance degradation, confirming that all components are crucial for improving the recommendation task. Notably, removing the alignment mechanism caused the most significant drop, highlighting its critical role. This is likely because entity embeddings $\mathbf{E}$ play a central role in the recommendation task, and the alignment module helps entities capture and integrate information more effectively, resulting in better representations for recommendations.

**Conversation Task.** We conducted similar ablation experiments for the conversation task, removing the knowledge tree-enhanced prompts, user preference prompts, and the information alignment module one by one in both the pretraining and training stages. Figure 4 shows that removing any component results in a performance drop, indicating that all components contribute positively to the conversation task. The most significant decline occurred when removing the tree structure, demonstrating its importance in the conversation task. This is because the tree structure provides rich entity-related information from the KG, which is effectively utilized by large language models to enhance conversational quality.

### 4.5 HYPERPARAMETER SENSITIVITY ANALYSES

**Analyses on the degree and depth of knowledge tree.** To study the influence of the size of the knowledge tree on the system, we conduct parameter-tuning experiments on both two datasets. We investigated the influence of the layer count (i.e., depth) and degree of knowledge trees extracted from the KG $G$ on the model's recommendation performance. As depicted in the figure5, both two-layer and one-layer trees yield similar best recommendation results. However, one-layer trees exhibit more stable performance. As the degree of the tree changes, the model's recommendation ability

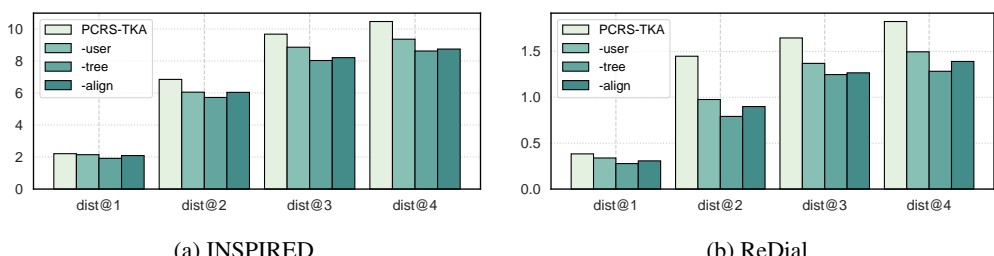

(a) INSPIRED                                      (b) ReDial

Figure 4: Ablation study on both two datasets about the conversation task. User and tree refer to two kinds of prompts. Align refers to the align loss in contrastive loss.

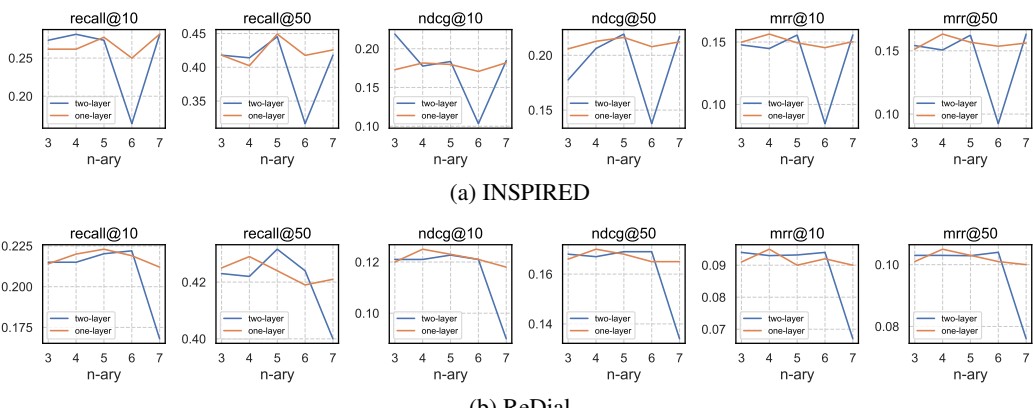

Figure 5: Model performance comparison with varying degrees and depth of the Knowledge tree in the conversation task. The x-axis represents the degree of the knowledge tree, and the legend indicates the depth of the tree.

remains relatively stable without major fluctuations. On the other hand, two-layer trees show a sharp decline in performance as the degree increases. This could be due to the more critical information contained in one-layer trees, and the model's limited capacity to process tree text length. Since the tree text is generated through depth-first traversal, increasing the tree depth may cause the text to focus only on a few deep relationships, leading to the loss of important entity information. Hence, it is advisable to personalize the tree's layer count and degree based on the structure of the KG and the frequency of entities appearing in dialogues.

**Analyses on the loss balancing.** As mentioned in 3.7, the loss function of the recommendation task contains two hyperparameters $\alpha$ and $\beta$, Alpha and beta are used to control the proportion of $L_{user}$ and $L_{align}$ in the loss function, i.e., the guiding degree for the model. We present the experimental results with varing $\alpha$ and $\beta$ in Figure 6 in the Appendix. We can observe that the optimal performance of alpha is around 0.02, while beta performs best at around 0.002. As the values increase, the recommendation ability of both will start to decrease. Compared to the influence of the user preference extraction module on entity embeddings, which is mainly through the fusion operation, $L_{align}$ has a greater impact on entity embeddings during training via alignment operations. If $L_{align}$ occupies too high a proportion during training, it may disrupt the information learned by entity through RGCN. On the other hand, a smaller proportion of $L_{align}$ will gradually guide the integration of entity embeddings and tree embeddings.

## 5 CONCLUSION

In this paper, we presented PCRS-TKA, a novel framework that integrates pretrained language models with KGs for conversational recommender systems. Our approach enhances the recommendation process through tree-structured knowledge augmentation and user preference extraction from multi-turn dialogues, improving both recommendation accuracy and conversational fluency. Additionally, the alignment module ensures effective integration of heterogeneous data from dialogues and KGs, minimizing semantic inconsistencies and noise. Finally, extensive comparative experiments on benchmark datasets clearly demonstrated the effectiveness of the proposed model compared to state-of-the-art baselines in terms of both recommendation quality and dialogue generation.

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

# A APPENDIX

## A.1 EXPERIMENTAL SETUP

Except for UniCRS, the baseline models are implemented using the CRSLab tool (Zhou et al., 2021). CRSLab is an open-source toolkit for building CRS, providing ready-made code where users only need to modify configuration files to run the systems in the tool. Furthermore, CRSLab has already set default values for model-specific parameters, such as the token hidden size of transformer module and the hidden size of RGCN module in KGSF, based on the code from the baseline repository papers, and has validated the correct selections of parameters on the TG-Redial dataset. So users only need to adjust some common parameters according to the dataset, including hidden size, number of training epochs, batch size, and optimizer configurations. For these parameters, we opted for a grid search approach, choosing training epochs from [5, 10, 20, 50], batch size from [8, 16, 32, 64, 128], and learning rates from [1e-4, 5e-4, 1e-3, 5e-3]. Both the recommendation and task modules use an early stopping strategy with the endurance epoch set to 3. After numerous experiments, we identified the best-performing parameter settings and uploaded the configuration file to the codebase. For the UniCRS task, we used the code and parameter configuration provided by the repository of UniCRS ar `https://github.com/rucaibox/unicrs`. We examined the evaluation metrics in all mentioned codebases and confirmed that the implementations of all evaluation metrics are identical. We eventually achieved results similar to those reported in the corresponding paper,

## A.2 HYPERPARAMETER SENSITIVITY ANALYSES

As mentioned in Section 4.5, we provide the figures for the hyperparameter sensitivity analyses on $\alpha$ and $\beta$.

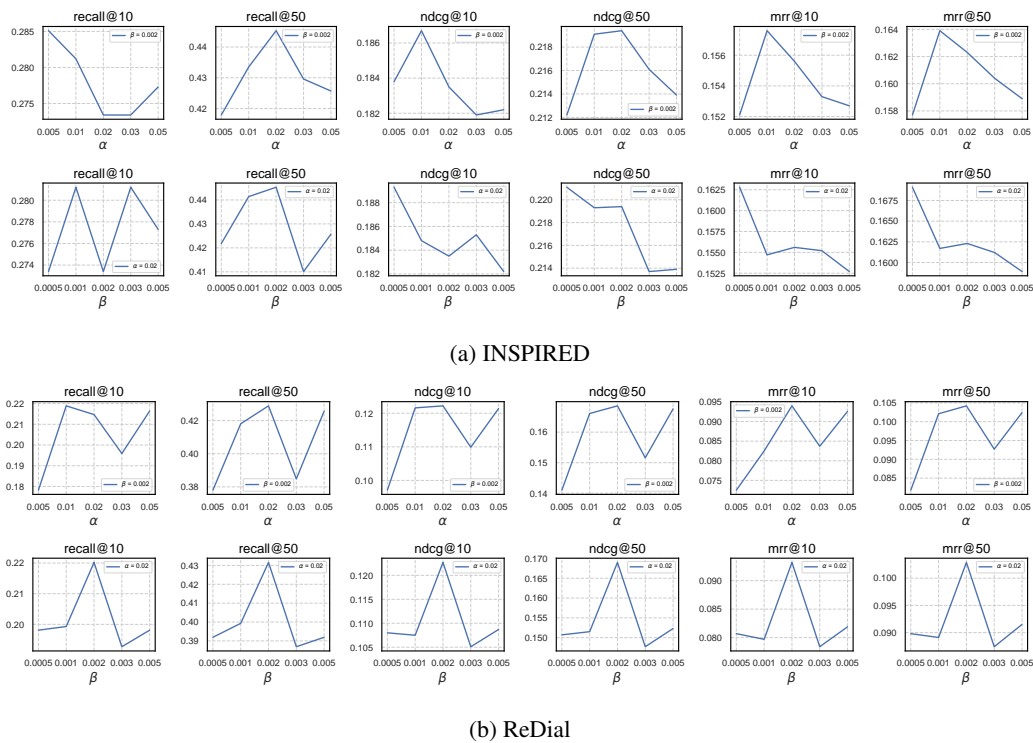

(a) INSPIRED

(b) ReDial

Figure 6: Model performance comparison with varing alpha and beta.

