# OpenReview forum: "Enhancing Conversational Recommender Systems with Tree-Structured Knowledge and Pretrained Language Models"
_ICLR.cc/2025/Conference — Submitted to ICLR 2025_

### Official Review · Reviewer_fzHg · 2024-10-28

**Soundness:** 2
**Presentation:** 3
**Contribution:** 2
**Rating:** 6
**Confidence:** 4

**Summary:**

Conversational recommender systems (CRS) enhanced by pretrained language models (PLMs) face challenges. The PCRS-TKA framework, integrating PLMs with knowledge graphs via prompt-based learning, addresses these by enhancing accuracy and personalization. Experiments show it outperforms existing methods in accuracy and fluency.

**Strengths:**

(1) This paper presented a signifcant extension on PLM based CRS with newly proposed method and extensive experiments.

(2) The results have shown improved performance on the task of conversational recommendation.

**Weaknesses:**

(1) The introduction could be improved by incorporating more movitation and rationale, e.g., how and why the proposed method would improve the performance?

(2) The proposed method seems somehow complicated. Can it be simplified to some extent? A simple yet efficient method is important in practice.

(3) Do you try larger language models except DialogGPT?

**Questions:**

See my comments in the weaknees

---

> ### Author Response · Authors · 2024-11-21
>
> We would like to thank you for your thoughtful and constructive feedback. Below, we address each of the points raised in your review:
> ### **Weakness 1**
> Thank you for your suggestion to enhance the motivation and rationale in the introduction. We will restructure the introduction to better highlight the innovative aspects of our method and how each component contributes to performance improvements. Specifically:
> 1. **Knowledge Tree Enhanced Module:** Current CRS models utilize RGCN to capture the relational information from KGs, and neglect the reasoning capability of PLMs for KG structure-based knowledges, so we introduced a Knowledge Tree Enhanced Module to use PLMs to capture the semantic relationships and structural information of KGs, providing background information for the recommendations.
> 2. **User Preference Extraction Module:** One key challenge in CRS is effectively capturing and utilizing user collaborative preference information, particularly from dialogue interactions. Traditional PLM-based methods often focus primarily on processing textual or contextual inputs, while overlooking the direct modeling of collaborative signals. To address this limitation, we have designed a User Preference Extraction Module that is jointly trained with the PLMs. This module captures collaborative preference signals directly from dialogue interactions, using them to guide the PLM's recommendation ability.
> 3. **Multimodal Information Integration:** Our model consists of multiple modules, so we introduce a more comprehensive Multimodal Information Integration to address the semantic inconsistency problem between features extracted by different modules.
>
> These three modules, working together, significantly improve the performance of the model in generating more accurate and contextually relevant responses and recommendations.
> ### **Weakness 2**
> Thank you for your comment about the complexity of the proposed method. While the model incorporates multiple modules to process different types of information, we want to clarify that the computational complexity of our approach is not significantly higher than that of existing methods. To clarify, we conducted a computational complexity analysis of PCRS-TKA, which is outlined as follows:
> * Feature Extraction Module: O(C² + M×R + N×R)
> * Knowledge Tree Enhanced Module: O(T²)
> * User Preference Extraction Module: O((C+E)²)
> * Multimodal Information Integration: O(T² + E²)
> * Prompt Learning Module: O(C² + I)
>
> Where:
> * C = min(max token length of PLM, max token length of conversations in a batch)
> * T = min(max token length of PLM, max token length of knowledge tree texts in a batch)
> * N = number of nodes in KG
> * R = type of edges in KG
> * M = number of edges in KG
> * E = number of entities mentioned in the conversation
> * I = number of candidate items
>
> For M>N>I and E<<C≈T, the total complexity of PCRS-TKA can be expressed as O(M×R + C²), which is comparable to the complexity of the baseline model, UniCRS, which also scales as O(M×R + C²). Thus, while PCRS-TKA incorporates additional modules, the overall computational cost remains competitive with baseline models.
> ### **Weakness 3**
> Thank you for your suggestion regarding the exploration of larger language models beyond DialoGPT. Despite LLMs like Llama-3-instruct may offer better performance due to their larger parameter sizes and enhanced reasoning abilities, but we focus more on PLMs for their suitability for full-scale fine-tuning and practical deployment in real-world scenarios, where computational efficiency and scalability are key considerations. According to your advice, to prove that PCRS-TKA can be flexibly transplanted to different PLMs, we conducted experiments replacing RoBERTa with BERT and replacing DialoGPT with GPT-2. The results (shown in the table below) demonstrate that PCRS-TKA performs better than UniCRS across different PLMs, and the choice of RoBERTa and DialoGPT is the most suitable for the specific task and dataset.
> | Model     | recall@10 | recall@50 | ndcg@10 | ndcg@50 | mrr@10 | mrr@50 |
> |-----------|-----------|-----------|---------|---------|--------|--------|
> | UniCRS(BERT)      | 0.250      | 0.426    | 0.162  | 0.201  | 0.138 | 0.143 |
> | PCRS-TKA(BERT)   | 0.254    | 0.441    | 0.174  | 0.214  | 0.148 | 0.156 |
> | UniCRS(GPT2)     | 0.230    | 0.422    | 0.153   | 0.196  | 0.129 | 0.138 |
> | PCRS-TKA(GPT2) | 0.258    | 0.430    | 0.162  | 0.204  | 0.140 | 0.147 |
> | UniCRS(origin)     | 0.262    | 0.406    | 0.159   | 0.193  | 0.131 | 0.138 |
> | PCRS-TKA(origin) | 0.273    | 0.445    | 0.184  | 0.220  | 0.156 | 0.162 |
>
> Once again, thank you for your valuable suggestions. We will incorporate these improvements and update the experimental results in the next revision.

---

### Official Review · Reviewer_i7PS · 2024-10-29

**Soundness:** 2
**Presentation:** 2
**Contribution:** 1
**Rating:** 3
**Confidence:** 5

**Summary:**

In this paper, authors focus on challenges included in PLMs and propose the PCRS-TKA framework, which integrates pretrained language models (PLMs) pretrained language models (PLMs) with knowledge graphs (KGs) using prompt-based learning to enhance conversational recommender systems (CRS).
By leveraging tree-structured knowledge from KGs, PCRS-TKA aims to improve the accuracy and reliability of recommendations while addressing challenges such as hallucinations and the need for personalized interactions.
This framework includes a user preference extraction module and an alignment module to ensure semantic consistency. Experimental results indicate that PCRS-TKA outperforms existing methods in both recommendation accuracy and conversational fluency.

**Strengths:**

1. **Integration of PLMs with KGs**:
   - The framework effectively combines the strengths of PLMs and KGs, enabling richer contextual understanding and improved recommendation accuracy.

2. **Reproducibility**:
   - The availability of the source code enhances the reproducibility of the results, allowing other researchers to validate and build upon the findings.

3. **Experimental Results**:
   - Extensive experiments demonstrate significant improvements in recommendation accuracy and conversational fluency, providing empirical support for the proposed framework.

4. **Personalization**:
   - The inclusion of a user preference extraction module shows a thoughtful approach to improving the personalization of recommendations, which is critical in conversational settings.

**Weaknesses:**

1. **Lack of Novelty**:
   - The framework appears to be a straightforward extension of existing methods, lacking unique ideas or contributions in the proposed modules, architecture, and training objectives.

2. **Insufficient Literature Review**:
   - The paper does not adequately survey recent advancements in prompt learning, which diminishes the contextual grounding of the proposed approach within the broader research landscape.

3. **Hallucination Challenge**:
   - While the authors mention hallucinations, they do not provide a concrete solution to this issue or conduct experiments to compare models concerning this challenge, limiting the impact of their claim.

4. **Generalizability Concerns**:
   - The choice of RoBERTa and DialoGPT as PLMs raises questions about the generalizability of PCRS-TKA. The framework's effectiveness with more recent models, such as LLama-3-instruct, remains unexplored.

5. **Limited Discussion on PLM Variability**:
   - There is a lack of discussion regarding the applicability of PCRS-TKA across diverse PLMs, as well as the potential advantages and disadvantages of different PLMs in the context of the framework.

**Questions:**

1. **Novelty and Unique Contributions**:
   - Can you elaborate on the unique contributions of PCRS-TKA compared to existing frameworks? How does it differ fundamentally from other prompt-based learning approaches?

2. **Addressing Hallucinations**:
   - Could you clarify how your framework specifically addresses the issue of hallucinations? Are there potential strategies you envision for mitigating this problem?

3. **Generalizability Across PLMs**:
   - Have you considered testing PCRS-TKA with state-of-the-art models like LLama-3-instruct? What are your thoughts on how the framework might perform with different PLMs?

4. **Discussion on Prompt Learning**:
   - How do you plan to integrate the latest findings in prompt learning into your framework? What implications might this have for your proposed methods?

5. **Future Research Directions**:
   - What future research directions do you foresee for enhancing the PCRS-TKA framework? Are there specific areas you believe warrant further exploration?

---

> ### Author Response · Authors · 2024-11-21
>
> Thank you for your insightful comments.
> ### **Weakness1 & Question 1**
> Here we will clarify several key innovations and improvements that distinguish our approach from existing frameworks:
> 1. **Knowledge Tree Enhanced Module:** We introduced a Knowledge Tree Enhanced Module that leverages PLMs to mine the structural information from KGs. This module extracts deeper semantic relationships and contextual knowledge, which enhances the recommendation quality.
> 2. **User Preference Extraction Module:** We introduced a User Preference Extraction Module that is jointly trained with the PLMs in an end-to-end manner. This module captures collaborative preference signals from dialogue interactions, which are incorporated as a user feedback loss during training. By jointly optimizing the PLM with this feedback loss, the recommendation model is guided to better reflect user-specific needs and improve the quality of personalized recommendations.
> 3. **Multimodal Information Integration:** Since our model consists of multiple modules, we introduce a more comprehensive Multimodal Information Integration to address the semantic inconsistency problem between features extracted by different modules.
>
> These innovations contribute to PCRS-TKA's superior performance compared to previous approaches and are integral to the model’s effectiveness.
> ### **Weakness 3 & Question 2**
> We appreciate your comments on the hallucination challenge. We specifically address hallucinations by deeply integrating knowledge graphs (KGs), which provide factual, structured knowledge that the model can rely on when generating responses. By grounding the model’s outputs in verified and relevant knowledge, we reduce the risk of generating speculative or incorrect content. To evaluate the effectiveness of our approach, we conducted a human evaluation experiment that included a new metric: “Information Inaccuracy”. Annotators assessed the factual accuracy and logical consistency of model outputs. As shown below, the results demonstrate that our framework reduces hallucinations more effectively than the baseline models:
> | Model    | Information Inaccuracy|
> |----------|----------------|
> | UniCRS   | 2.71            |
> | PCRS-TKA | 2.53            |

---

> > ### Author Response · Authors · 2024-11-21
> >
> > ### **Weakness 4 & Question 3**
> > Thank you for your comment regarding the generalizability of PCRS-TKA across different PLMs. While we chose RoBERTa and DialoGPT based on their suitability for the specific task and dataset, we recognize that larger models like LLaMA-3-instruct may offer superior performance due to their enhanced reasoning abilities. However, we opted for PLMs in our study because of their efficiency and flexibility in fine-tuning. To demonstrate PCRS-TKA’s flexibility with different PLMs, we tested the framework with BERT and GPT-2, replacing RoBERTa and DialoGPT. The results, as shown below, indicate that PCRS-TKA outperforms UniCRS across these PLMs, confirming that our approach is adaptable to different model architectures. This demonstrates that the choice of PLMs does not significantly undermine the flexibility of PCRS-TKA, and it can be easily adapted to a range of PLMs.
> > | Model     | recall@10 | recall@50 | ndcg@10 | ndcg@50 | mrr@10 | mrr@50 |
> > |-----------|-----------|-----------|---------|---------|--------|--------|
> > | UniCRS(BERT)      | 0.250      | 0.426    | 0.162  | 0.201  | 0.138 | 0.143 |
> > | PCRS-TKA(BERT)   | 0.254    | 0.441    | 0.174  | 0.214  | 0.148 | 0.156 |
> > | UniCRS(GPT2)     | 0.230    | 0.422    | 0.153   | 0.196  | 0.129 | 0.138 |
> > | PCRS-TKA(GPT2) | 0.258    | 0.430    | 0.162  | 0.204  | 0.140 | 0.147 |
> > | UniCRS(origin)     | 0.262    | 0.406    | 0.159   | 0.193  | 0.131 | 0.138 |
> > | PCRS-TKA(origin) | 0.273    | 0.445    | 0.184  | 0.220  | 0.156 | 0.162 |
> > ### **Weakness 2 & Question 4**
> > Thank you for pointing out the gaps in the literature review. We will revise the paper to include a more thorough survey of recent advancements in prompt learning. Specifically, we will discuss the evolving techniques in prompt-based learning, including the work on retrieval-augmented generation (RAG) and meta-learning approaches, which have shown promise in improving the flexibility and performance of PLMs in dialogue systems. Incorporating these insights will help better contextualize PCRS-TKA within the broader research landscape. Our PCRS-TKA framework is designed as a general and flexible architecture, allowing for the integration of various prompt learning approaches to further improve its performance and efficiency. For instance, techniques such as dynamic prompt generation and prompt engineering can be incorporated to fine-tune the model for specific tasks, ensuring better alignment with conversational contexts and further improving recommendation quality. We will conduct more studies on this direction for the future research.
> > ### **Question 5**
> > Looking ahead, we aim to enhance the PCRS-TKA framework through several key avenues:
> > * **Integration of Advanced Prompt Learning Techniques:** As you noted, incorporating new prompt learning methods offers the potential for improving the framework's flexibility and performance. For instance, dynamic prompt generation, retrieval-augmented prompt refinement, or task-specific prompt engineering could be explored to better adapt the framework to varying conversational contexts and improve knowledge utilization.
> > * **Enhancing User Feedback Utilization:** A key area for future exploration is improving how user feedback signals are captured and applied. By leveraging techniques like adaptive feedback loops or reinforcement learning, the framework could dynamically adjust its recommendations based on evolving user preferences, enabling more precise personalization over time.
> >
> > Once again, thank you for your valuable suggestions. We will incorporate these improvements and update the experimental results in the next revision.

---

> > > ### Comment · Reviewer_i7PS · 2024-11-22
> > > **Thank you for official comment**
> > >
> > > I appreciate your comments and will review your other comments.

---

### Official Review · Reviewer_1Hk2 · 2024-10-29

**Soundness:** 2
**Presentation:** 2
**Contribution:** 1
**Rating:** 3
**Confidence:** 5

**Summary:**

This paper proposes PCRS-TKA, a framework that enhances conversational recommendation systems by integrating pre-trained language models with knowledge graphs through knowledge-enhanced prompt learning.
The framework consists of four main components: a feature encoder module for encoding dialogue and graph information, a knowledge tree module for KG triple transformation, a user preference module for preference extraction, and a soft-prompt module for task guidance. The combined prompt template is then used by DialoGPT to generate conversational recommendations.

**Strengths:**

- Clear presentation of the framework and results.
- Interesting perspective on representing KG triples as a tree structure for prompt learning for CRS.

**Weaknesses:**

- PCRS-TKA heavily relies on established techniques and follows UniCRS[1] architecture, While the knowledge tree module introduces some innovation, other components like constructive learning and prompt-based design are mainly adaptations of existing methods from CRS [1][2].
- The choice of PLMs needs justification given LLMs have demonstrated promising performance in CRS.
- The benchmark comparison focuses on older models (up to 2022); the authors should include later models.
- The authors emphasize PLM's logical reasoning abilities and KG for hallucination reduction, there's no systematic evaluation of these. Also, the human evaluation survey from the provided link actually shows many responses contain incorrect facts or hallucinations

[1] Wang, X., Zhou, K., Wen, J.-R., Zhao, W.X., 2022. Towards Unified Conversational Recommender Systems via Knowledge-Enhanced Prompt Learning, in: Proceedings of the 28th ACM SIGKDD Conference on Knowledge Discovery and Data Mining. Presented at the KDD ’22: The 28th ACM SIGKDD Conference on Knowledge Discovery and Data Mining, ACM, Washington DC USA, pp. 1929–1937. https://doi.org/10.1145/3534678.3539382
[2] Zhou, Y., Zhou, K., Zhao, W.X., Wang, C., Jiang, P., Hu, H., 2022. C2-CRS: Coarse-to-Fine Contrastive Learning for Conversational Recommender System, in: Proceedings of the Fifteenth ACM International Conference on Web Search and Data Mining. Presented at the WSDM ’22: The Fifteenth ACM International Conference on Web Search and Data Mining, ACM, Virtual Event AZ USA, pp. 1488–1496. https://doi.org/10.1145/3488560.3498514.
[3] Yang, B., Han, C., Li, Y., Zuo, L., Yu, Z., 2022. Improving Conversational Recommendation Systems’ Quality with Context-Aware Item Meta-Information, in: Carpuat, M., de Marneffe, M.-C., Meza Ruiz, I.V. (Eds.), Findings of the Association for Computational Linguistics: NAACL 2022. Association for Computational Linguistics, Seattle, United States, pp. 38–48. https://doi.org/10.18653/v1/2022.findings-naacl.4

**Questions:**

* In line 415, the authors attribute PCRS-TKA's higher performance on INSPIRED to "more dialogues", yet INSPIRED actually contains fewer dialogues than ReDial. Could the authors clarify this contradiction and provide a more accurate analysis of the performance gains?

2. How is the tree-structured knowledge representation significantly different from MESE’s[3] approach of using item metadata as knowledge prompts, especially when the knowledge tree is set to hop=1? And what specific advantages does this added complexity bring over simpler metadata-based methods?

---

> ### Author Response · Authors · 2024-11-21
> **part1**
>
> Thank you for your insightful comments.
>
> ### **Weakness 1**
> Thanks for your comments. We would like to highlight several key innovations in our approach that distinguish it from prior work, particularly UniCRS [1].
>
> 1. **Knowledge Tree Enhanced Module:** As you noted, we introduced a Knowledge Tree Enhanced Module that leverages PLMs to mine the structural information from KGs. This module extracts deeper semantic relationships and contextual knowledge, which enhances the recommendation quality.
> 2. **User Preference Extraction Module:** We have introduced a User Preference Extraction Module that is jointly trained with the PLMs in an end-to-end manner. This module captures collaborative preference signals from dialogue interactions, which are incorporated as a user feedback loss during training. By jointly optimizing the PLM with this feedback loss, the recommendation model is guided to better reflect user-specific needs and improve the quality of personalized recommendations.
> 3. **Multimodal Information Integration:** Since our model consists of multiple modules, we introduce a more comprehensive Multimodal Information Integration to address the semantic inconsistency problem between features extracted by different modules.
>
> These innovations contribute to PCRS-TKA's superior performance compared to previous approaches and are integral to the model’s effectiveness.
>
> ### **Weakness 2**
> Thank you for your comments regarding the choice of PLMs over LLMs. While it is true that LLMs have demonstrated superior performance in many tasks due to their larger parameter sizes and ability to capture complex relationships, PLMs are more suitable for full-scale fine-tuning and real-world deployment, particularly when practical resources are limited. Both PLMs and LLMs have their merits, and our work focuses on PLMs to provide a more efficient and deployable solution for real-world applications.
>
> ### **Weakness 3**
> We appreciate your suggestion to include a broader range of recent models in our comparison. In response, we have incorporated three state-of-the-art models [2, 3, 4] into our evaluation. The updated results demonstrate that PCRS-TKA outperforms most other models. This improvement can be attributed to the superior integration of knowledge graph (KG) information and the incorporation of the proposed User Preference Extraction Module, which captures collaborative preference signals from dialogue interactions. The updated comparison table is as follows:
> | Model      | recall@10 | recall@50 | ndcg@10 | ndcg@50 | mrr@10| mrr@50| dist2 | dist3 | dist4 |
> |------------|-----------|-----------|---------|---------|-------|-------|-------|---|----|
> | DCRS[2]    | 0.226  | 0.414   | 0.153   | 0.192  |0.130 |0.137 | 3.95  | 5.73  | 6.23  |
> | ReFICR[3]   | 0.274  | 0.396   | -       | -  | - | - | 4.111  | -  | 6.17   |
> | MPKE[4]    | 0.262  | 0.445   | -  | -   | 0.115   | 0.127   | 3.392  | 5.189  | 6.177   |
> | PCRS-TKA  | 0.273  | 0.445  | 0.184   | 0.22   | 0.156   | 0.162    | 6.85  | 9.67  | 10.46 |
>
> ### **References**
> [1] Wang, Xiaolei, et al. "Towards unified conversational recommender systems via knowledge-enhanced prompt learning." Proceedings of the 28th ACM SIGKDD Conference on Knowledge Discovery and Data Mining. 2022.
>
> [2] Dao, Huy, et al. "Broadening the view: Demonstration-augmented prompt learning for conversational recommendation." Proceedings of the 47th International ACM SIGIR Conference on Research and Development in Information Retrieval. 2024.
>
> [3] Yang, Ting, and Li Chen. "Unleashing the Retrieval Potential of Large Language Models in Conversational Recommender Systems." Proceedings of the 18th ACM Conference on Recommender Systems. 2024.
>
> [4] Zhang, Chengyang, et al. "Improving conversational recommender systems via multi-preference modelling and knowledge-enhanced." Knowledge-Based Systems 286 (2024): 111361.

---

> > ### Author Response · Authors · 2024-11-21
> > **part2**
> >
> > ### **Weakness 4**
> > We appreciate your comments regarding the hallucination problem. To better evaluate how our model mitigates hallucinations, we conducted a human evaluation experiment. Specifically, we introduced a metric called "Information Inaccuracy" to evaluate the factual accuracy and logical consistency of model responses, lower values indicate better performance in reducing hallucinations. Then we added it to the questionnaire mentioned in our paper and invited three annotators to score the responses generated by models. As shown below, the results demonstrate that our framework reduces hallucinations more effectively than the baseline models. We will add more discussions for revisions according to your comments.
> > | Model    | Information Inaccuracy |
> > |----------|----------------|
> > | UniCRS   | 2.71            |
> > | PCRS-TKA | 2.53            |
> >
> > ### **Question 1**
> > Sorry for the confusion in our original statement. To clarify, the higher performance of PCRS-TKA on the INSPIRED dataset is not due to the total number of dialogues but rather the greater average number of dialogue turns within each conversation in INSPIRED compared to ReDial. This difference allows the model to capture more complex user preference features over the course of longer conversations, which contributes to the improved performance we observed. We have provided a table below comparing the average number of turns in each dataset to clarify this:
> > | Dataset   | Conversations |Avg. Utterances/Conv |Avg. Words/Utterance |
> > |-----------|-------|-----------------|-----------------|
> > | ReDial    | 10006 | 18.20408        | 14.5            |
> > | INSPIRED  | 1001  | 35.77522        | 19              |
> >
> > ### **Question 2**
> > We appreciate your comparison with the MESE method. While both our knowledge tree and MESE’s item metadata-based approach utilize external knowledge, and at hop=1, the structure of knowledge tree text may appear similar to MESE, but the key distinction lies in the structural information captured by our Knowledge Tree Enhanced Module. Unlike metadata, which offers unidimensional, flat information, our Knowledge Tree Enhanced Module captures tree-based relationships between entities in the KG. Additionally, our method does not introduce significant computational complexity compared to metadata-based methods. Both approaches have a complexity of O(n²), where n denotes the token length of PLM's input and n grows linearly with the number of entities, which allows for deeper knowledge extraction.
> >
> > We hope that these clarifications address your concerns. Thank you again for your insightful feedback. We look forward to incorporating these improvements in the revised manuscript.

---

> > > ### Comment · Reviewer_1Hk2 · 2024-11-27
> > >
> > > Thanks to the authors for the responses and for providing additional experiments. However, only the results on INSPIRED are reported. The performance of DCRS and ReFICR on ReDial is better than that of the proposed method.

---

> > > > ### Author Response · Authors · 2024-11-27
> > > >
> > > > Thank you for your valuable feedback. Due to time constraints, we were unable to evaluate the performance of the new baselines on the ReDial dataset.
> > > >
> > > > However, based on the results presented in the relevant papers, DCRS, by incorporating an external retrieval library, demonstrates improved recommendation performance, slightly surpassing PCRS-TKA on the ReDial dataset. On the other hand, the ReFICR method, which utilizes the GRITLM-7B model, significantly outperforms all non-LLM approaches on the same dataset. That being said, we would like to emphasize that the computational resource requirements of large models are significantly higher than those of our proposed method.
> > > >
> > > > Despite this, both DCRS and ReFICR perform considerably worse than our model on the Inspired dataset, suggesting that our approach achieves comparable performance to LLM-based methods while consuming much fewer resources. This makes our method particularly advantageous in practical environments, where local training and deployment are essential.
> > > >
> > > > Furthermore, we believe that non-LLM recommendation systems still hold significant practical value, especially in scenarios where resource efficiency is crucial. We kindly ask you to consider this aspect.
> > > >
> > > > Thank you for your understanding and look forward to your further response.

---

### Official Review · Reviewer_39A5 · 2024-11-03

**Soundness:** 3
**Presentation:** 3
**Contribution:** 2
**Rating:** 5
**Confidence:** 5

**Summary:**

This paper addresses two key limitations of existing Conversational Recommender Systems (CRS): (1) hallucinations and (2) difficulties in providing precise, entity-specific recommendations.
To overcome these challenges, this paper proposes the PCRS-TKA framework. This framework is designed to enhance the accuracy and reliability of CRS by effectively aligning tree-structured information from knowledge graphs with the semantic information in conversation history through prompt learning.

**Strengths:**

- The authors clearly and thoroughly explained the proposed method to address the limitations of Conversational Recommender Systems (CRS).
- The authors demonstrate the effectiveness of the proposed framework and its components in enhancing recommendation capability.

**Weaknesses:**

- **The proposed method does not fully handle the limitations.**
The authors pointed out the issue of inaccurate generation caused by hallucinations. However, the proposed framework has different input prompts for the recommendation and response generation tasks. While the proposed module components can effectively bridge the semantic gap between the conversation prompt and the entity prompt, they cannot completely resolve the issue. Providing different input prompts to the same model can lead to additional semantic misalignment issues, as the prompts may cause the model to perform distinct tasks even if they are connected through a semantic alignment process. This could lead to the fundamental issue of semantic inconsistency between recommendations and conversations, as highlighted in previous research[1]. Therefore, additional experimental or statistical evidence is needed. For instance, it is necessary to examine how consistently the generated dialogue aligns with the predicted recommended items and the related entities from the knowledge graph to effectively address the hallucination problem.


- **The conducted dataset exhibits a "repeated item shortcut" problem, which refers to data leakage where items have appeared in previous conversation turns [2][3].**
More than 15% ground-truth items are repeated items in INSPIRED dataset. This issue suggests that the proposed structure utilizes the embedding of the ground truth (GT) item as the input prompt. Additional validation is needed to assess whether the proposed method remains effective in scenarios eliminating repeated items as ground truth.

[1] Wang, Xiaolei, et al. "Towards unified conversational recommender systems via knowledge-enhanced prompt learning." Proceedings of the 28th ACM SIGKDD Conference on Knowledge Discovery and Data Mining. 2022.

[2] He, Zhankui, et al. "Large language models as zero-shot conversational recommenders." Proceedings of the 32nd ACM international conference on information and knowledge management. 2023.

[3] Zhu, Lixi, Xiaowen Huang, and Jitao Sang. "How Reliable is Your Simulator? Analysis on the Limitations of Current LLM-based User Simulators for Conversational Recommendation." Companion Proceedings of the ACM on Web Conference 2024. 2024.

**Questions:**

- I agree that the proposed framework effectively handles the structured information of the KG and text semantic information through various fusion modules. But I believe further analysis of the two well-known problems mentioned above is necessary.
- The results of the analyses on the degree and depth of the knowledge tree in Section 4.5 are intriguing. I agree that an increase in depth can introduce noisy information and may lead to the loss of information regarding the nearest relationships. However, could it also be interpreted that the embedding model fails to adequately reflect the structural information of the proposed construction of the knowledge tree text? I wonder about the potential outcomes if the knowledge tree text is better structured or if a more powerful embedding model is used.

---

> ### Author Response · Authors · 2024-11-21
>
> Thank you for your insightful comments. We appreciate the opportunity to further clarify and improve our work.
> ### **Weakness 1 & Question 1**
> Thank you for your insightful suggestion. As noted, in our initial approach, we followed the settings in [1], where the responses generated by the conversation task were used as part of the prompts for the recommendation task. Inspired by your comment, we conducted additional experiments where we unified the prompt for both tasks, using the recommendation task prompt ($P_{rec}$) as the input for both. The results, presented in the table below, demonstrate that this adjustment not only addresses the semantic misalignment issue but also unexpectedly enhances the performance of the conversation task. We found this approach to be highly effective, and we will include a detailed discussion of these findings in the revised manuscript. Thank you for helping us uncover this valuable improvement.
> | Model                    | dist1  | dist2  | dist3  | dist4  |
> |--------------------------|--------|--------|--------|--------|
> | PCRS-TKA                 | 2.209  | 6.851  | 9.676  | 10.465 |
> | PCRS-TKA($P_{rec}$ for both tasks)| 2.347  | 6.805  | 9.906  | 10.804 |
>
> Additionally, we appreciate your comments on the hallucination challenge. To evaluate the effectiveness of our approach in mitigating hallucinations, we conducted a human evaluation experiment that included a new metric: “Information Inaccuracy”. Annotators assessed the factual accuracy and logical consistency of model outputs, lower values indicate better performance in reducing hallucinations. As shown below, the results demonstrate that our framework reduces hallucinations more effectively than the baseline models. We will add more discussions for revisions according to your comments.
> | Model    | Information Inaccuracy |
> |----------|----------------|
> | UniCRS   | 2.71            |
> | PCRS-TKA | 2.53            |
>
> ### **Weakness 2 & Question 1**
> We also appreciate your comments on the "repeated item shortcut" problem. After reviewing your suggestions, we adopted the experimental setup from [2] and carried out separate evaluations for repeated items and new items. As shown in the results below, we found that while the PCRS-TKA model does exhibit the "repeated item shortcut" issue, it still outperforms the baseline models for both repeated and new items. These findings are important for understanding the robustness of our method in the presence of such data leakage, and we will incorporate these results into the revised manuscript.
> | Model               | recall@10 | recall@50 | ndcg@10 | ndcg@50 | mrr@10 | mrr@50 |
> |------------------|-----------|-----------|---------|---------|--------|--------|
> | UniCRS(new)            | 0.1643    | 0.3333    | 0.104   | 0.1276  | 0.0758 | 0.0798 |
> | UniCRS(repeated)        | 0.5348    | 0.6976    | 0.4415  | 0.4734  | 0.4102 | 0.4175 |
> | PCRS-TKA(new)          | 0.1784    | 0.3427    | 0.1073  | 0.1434  | 0.0905 | 0.0981 |
> | PCRS-TKA(repeated)      | 0.6511    | 0.7411    | 0.5128  | 0.5331  | 0.4697 | 0.4739 |
>
> ### **Question 2**
> Thank you for raising the important question regarding the impact of knowledge tree depth and the embedding model on performance. We have conducted experiments with more complex embedding models, such as T-GNN [3], and tried various knowledge tree text structures containing different depths. However, our results indicate that neither the use of a more complex embedding model nor a different structuring of the knowledge tree text resulted in significant improved performance. This aligns with findings in the broader literature on knowledge graphs (KGs), which highlight that multi-hop connections often introduce additional noise that can outweigh their potential benefits [4]. According to your comments, we will add more discussions for revision.
>
> Once again, we thank you for your valuable feedback, which has helped us improve our work. We look forward to incorporating these changes and discussing the results in the updated manuscript.
>
> ### **References**
> [1] Wang, Xiaolei, et al. "Towards unified conversational recommender systems via knowledge-enhanced prompt learning." Proceedings of the 28th ACM SIGKDD Conference on Knowledge Discovery and Data Mining. 2022.
>
> [2] He, Zhankui, et al. "Large language models as zero-shot conversational recommenders." Proceedings of the 32nd ACM international conference on information and knowledge management. 2023.
>
> [3] Qiao, Z., Wang, P., Fu, Y., Du, Y., Wang, P., & Zhou, Y. (2020). Tree Structure-Aware Graph Representation Learning via Integrated Hierarchical Aggregation and Relational Metric Learning.
>
> [4] Jay Pujara, Eriq Augustine, and Lise Getoor. 2017. Sparsity and Noise: Where KG Embeddings Fall Short. In Proceedings of the 2017 Conference on Empirical Methods in Natural Language Processing, pages 1751–1756, Copenhagen, Denmark. Association for Computational Linguistics.

---

> > ### Comment · Reviewer_39A5 · 2024-11-26
> >
> > Thank you to the authors for the detailed responses. After carefully reviewing your explanations, I have decided to maintain my original score.

---

### Author Response · Authors · 2024-11-26
**Kindly Seeking Confirmation on Review Feedback for ICLR 2025**

Dear Reviewers,

We sincerely appreciate your time and effort in reviewing our manuscript and providing valuable feedback.

As the review discussion phase for ICLR 2025 draws to a close, we would like to kindly confirm whether our responses to your comments have effectively addressed your concerns. We submitted detailed replies to your feedback a few days ago, and we hope they have adequately clarified the issues raised.

If there are any remaining questions or additional points you would like us to address, please do not hesitate to let us know. We are more than willing to provide further clarification or engage in further discussion to ensure all your concerns are resolved.

Best regards,
The Authors

---

### Meta-Review · Area_Chair_GSMZ · 2024-12-24

**Metareview:**

This paper studies the conversational recommendation problem. The authors propose a new framework, named PCRS-TKA, which integrates pre-trained language models with knowledge graphs for conversational recommendation through prompt-based learning. The authors have performed extensive experiments to demonstrate the effectiveness of the proposed method, in terms of both recommendation accuracy and conversational fluency.

Overall, this paper introduces some new ideas about conversational recommendation. However, the technical novelty of the proposed method is limited. The experimental evaluation is not sufficient. More SOTA baseline methods should be included. The literature review is not sufficient. In addition, there is no systematic evaluation of the PLM's logical reasoning abilities and KG for hallucination reduction.

**Additional Comments On Reviewer Discussion:**

In the rebuttal, the authors provide more experimental results studying the impacts of the prompt for both tasks, the experimental results for repeated items and new items, the experimental results of baseline methods DCRS, ReFICR, MPKE. They also discuss the complexity of the proposed method, which is competitive with existing baseline methods. In addition, the authors also provide experimental results showing the performance with respect to different PLMs.

---

### Decision · Program_Chairs · 2025-01-22

Reject